# Optimization of Ultrasound-Assisted Extraction of Bioactive Compounds from *Pelvetia canaliculata* to Sunflower Oil

**DOI:** 10.3390/foods10081732

**Published:** 2021-07-27

**Authors:** Gabriela Sousa, Marija Trifunovska, Madalena Antunes, Isabel Miranda, Margarida Moldão, Vítor Alves, Rajko Vidrih, Paula Allen Lopes, Luis Aparicio, Marta Neves, Carla Tecelão, Suzana Ferreira-Dias

**Affiliations:** 1LEAF—Linking Landscape, Environment, Agriculture and Food—Research Center, Instituto Superior de Agronomia, Universidade de Lisboa, 1349-017 Lisbon, Portugal; gabriela.spsousa@gmail.com (G.S.); trifunovska93@hotmail.com (M.T.); mmoldao@isa.ulisboa.pt (M.M.); vitoralves@isa.ulisboa.pt (V.A.); 2Biotechnical Faculty, University of Ljubljana, 1000 Ljubljana, Slovenia; rajko.vidrih@bf.uni-lj.si; 3MARE—Marine and Environmental Sciences Centre, Politécnico de Leiria, 2520-641 Peniche, Portugal; madalena.f.antunes@ipleiria.pt (M.A.); marta.neves@ipleiria.pt (M.N.); carla.tecelao@ipleiria.pt (C.T.); 4Centro de Estudos Florestais, Instituto Superior de Agronomia, Universidade de Lisboa, 1349-017 Lisbon, Portugal; imiranda@isa.ulisboa.pt; 5Sovena Group, 1495-131 Algés, Portugal; paula.allen.lopes@sovena.pt (P.A.L.); Luis.aparicio@sovena.es (L.A.)

**Keywords:** algae, antioxidants, bioactive compounds, oxidative stability, *Pelvetia canaliculata*, response surface methodology, sunflower oil, ultrasound-assisted extraction

## Abstract

In this study, *Pelvetia canaliculata* L. macroalga, collected from the Atlantic Portuguese coast, was used as a source of bioactive compounds, mostly antioxidants, to incorporate them in sunflower oil with the aim of increasing its biological value and oxidative stability. The lyophilized alga was added to the oil, and ultrasound-assisted extraction (UAE) was performed. Algae concentration and UAE time varied following a central composite rotatable design (CCRD) to optimize extraction conditions. The following parameters were analyzed in the oils: oxidation products, acidity, color, chlorophyll pigments, carotenoids, flavonoids, total phenolic content, antioxidant activity by DPPH (2,2-diphenyl-1-picrylhydrazyl) and FRAP (ferric reducing antioxidant power) assays, and sensory analysis. Extraction conditions did not affect the acidity and the amount of oxidation products in the oil. Chlorophylls and carotenoid contents increased with algae concentration, while flavonoid extraction did not depend on algae content or UAE time. Total phenolics in the oil were highly related only to FRAP antioxidant activity. Storage experiments of supplemented oil (12.5% algae; 20 min UAE) were carried out under accelerated oxidation conditions at 60 °C/12 days. Antioxidant activity (FRAP) of supplemented oil was 6-fold higher than the value of non-supplemented oil. Final samples retained 40% of their initial antioxidant activity. The presence of algae extracts contributed to the increased oxidative stability of sunflower oil.

## 1. Introduction

Sunflower (*Helianthus annuus* L.) oil is obtained by physical or chemical (*n*-hexane) extraction of the sunflower seeds. This oil has wide culinary uses, i.e., for cooking, bakery, and frying. It is also used in the food industry for frying, production of emulsions, sauces, spreads, fillings, and in many different food formulations, namely margarine and mayonnaise.

The worldwide production of sunflower oil in 2018/2019 reached 19.34 million metric tons, which puts sunflower oil in fourth place of the most used oils in food, after palm, soybean, and rapeseed oils [1]. Therefore, the improvement of both the quality and stability of sunflower oil are central points for the food industry.

The limited oxidative stability of edible oils is one of the main constraints for the food industry, leading to a loss of nutritional value, a decrease in shelf-life, and the development of undesirable flavors. The resistance of edible oils to oxidation during processing, handling, and storage is related to their fatty acid composition, antioxidant content, and processing, storage, and handling conditions [2]. Sunflower oil is rich in polyunsaturated (PUFA) linoleic acid, C18:2 (48.3–74.0%) and monounsaturated (MUFA) oleic acid, C18:1 (14.0–39.4%) [3,4,5]. Since PUFA are more prone to oxidation, the high content of linoleic acid in sunflower oil will inevitably affect its oxidative stability [6].

To improve oxidative stability and to maintain the nutritional and sensory properties of refined oils, antioxidants are widely used in the food industry, following specific and tight legislation [3,7,8]. In addition to natural antioxidants, such as tocopherols, synthetic antioxidants, namely tertiary butyl hydroquinone (TBHQ), butylated hydroxyanisole (BHA), and butylated hydroxytoluene (BHT), can also be used [3].

Recently, the search for natural sources of antioxidants is increasing due to toxicological effects associated with synthetic ones [9]. Therefore, the supplementation of edible oils with natural antioxidants (namely from herbs, fruits, seeds, and processing by-products) has gained increasing interest in the last years [10,11,12].

Previous studies showed that the addition of natural antioxidants to sunflower oil improves its oxidative stability. In this context, rosemary (*Rosmarinus officinalis* L.) essential oil [13], *Salix aegyptiaca* L. extracts and essential oil [14], and *Pterospartum tridentatum* (L.) Willk. ethanolic extracts [15] were used to prevent the oxidation of sunflower oils.

Algae are known to be a source of valuable natural compounds, namely, polysaccharides, vitamins, proteins, steroids, dietary fibers, and natural antioxidants, which can be applied to cosmetics, food production, and food supplements [16,17,18,19,20]. *Pelvetia canaliculata* L. is a brown alga widely present on the Atlantic coast of Portugal [21]. This alga is rich in bioactive compounds, especially in phenolic compounds [22,23,24,25]. In addition, it has the advantage of lacking lignin, which makes easier the extraction of the interested biomolecules [26]. In the last years, the use of eco-friendly methods, such as ultrasound-assisted extraction (UAE), as an alternative to solvent extraction methods to recover bioactive compounds, is increasing [27,28].

The aim of this study was to enhance the nutritional quality and oxidative stability of refined sunflower oil by supplementation with bioactive compounds, predominantly antioxidants, from *P. canaliculata* directly to the oil by ultrasound-assisted extraction (UAE). The effect of UAE time and concentration of algae on oil oxidative stability, as well as on the content of bioactive compounds, were evaluated by response surface methodology (RSM), following a central composite rotatable design (CCDR). The oxidative stability of supplemented sunflower oil, obtained under selected UAE conditions, was also evaluated at 60 °C.

## 2. Materials and Methods

### 2.1. Materials

Refined sunflower oil without added antioxidants was kindly donated by SOVENA Portugal, Consumer Goods SA. The brown macroalga *Pelvetia canaliculata* L. was harvested at the beach of Pedras do Corgo, Portugal (41°14′55.52″ N, 8°43′29.89″ W) in June 2019. Gallic acid (97.5% purity), Trolox (97% purity), quercetin (95% purity), and catechin (98% purity) used for calibration curves were purchased from Sigma and Acros Organics. All the solvents and reagents used were analytical grade and obtained from different suppliers.

### 2.2. Characterization of Pelvetia canaliculata

Algae biomass was characterized regarding (i) proximate composition (total protein, lipids, carbohydrates, and ash), (ii) pigments (chlorophyll and carotenoid contents) in *n*-hexane extracts, (iii) total phenolic compounds and flavonoids in methanol extracts, and (iv) antioxidant activity in methanol extracts.

#### 2.2.1. Algae Material and Extracts Preparation

After harvest, algae material was cleaned from extraneous matter, frozen at −80 °C (Haier Biomedical, DW-86L728J, Qingdao, China), freeze-dried (Telstar, Lyoquest-85, Telstar Portugal, Lisbon, Portugal), ground to powder, and stored at −20 °C until use. The extracts were prepared by vortex stirring the freeze-dried algae with *n*-hexane (6.24%, *m*/*v*) or methanol (6.28%, *m*/*v*) for 6 min. Then, the suspension was centrifuged (10 min at 3000 rpm) to separate the solid residue. Finally, the extracts were stored, protected from light, in the refrigerator at 4 °C, until further analyses.

#### 2.2.2. Proximate Composition

Crude protein of freeze-dried *P. canaliculata* was determined by the Kjeldahl method, using the conversion factor of 5.38, which is specific for brown algae [29].

Total lipid content was analyzed according to Folch et al. [30] with slight modifications. Briefly, lyophilized algae samples (1 g) were hydrated with 1 mL of deionized water and extracted with 10 mL of Folch reagent (CHCl_3_:CH_3_OH, 1:2 *v*/*v*) under vortex stirring. After addition of 1.2 mL of NaCl 0.8% (*m*/*v*), samples were homogenized and centrifuged (5 min at 6000 rpm) for phase separation. The lower phase was recovered and filtered through an anhydrous sodium sulfate column to a flask previously weighed. Then, chloroform (5 mL) was added to the remaining sample material, and the extraction procedure was repeated. Finally, the solvent was removed in a rotary evaporator (Heidolph 2, LAB1ST, Shanghai, China), and the lipid extract was dried at 40 °C until constant weight. Ash quantification was carried out by incineration of lyophilized algae samples (500 mg) in a muffle (Nabertherm, Liliemthal/Bermen, Germany) at 500 °C for 12 h. Total carbohydrates were calculated by difference, considering protein, lipids, and ash values. All the results were expressed as mean ± STD, on a dry weight basis (% d.w.), of four independent analyses.

#### 2.2.3. Chlorophyll and Carotenoid Contents

The chlorophyll content of *n*-hexane extract of *Pelvetia canaliculata* was determined according to the method described by Pokorny et al. [31], with modifications, and expressed as mg pheophytin a/kg algae (d.w.). The extract was directly analyzed in a UV-Vis double beam spectrophotometer (Agilent Technologies cary series 100 UV-Vis, Santa Clara, CA, USA) by measuring the absorbance in a 5 mm cell at wavelengths: 630, 670, and 710 nm using air as reference. Three replicates of each sample were analyzed.

For quantification of carotenoids, the extract was diluted in *n*-hexane in a 1:1 ratio (*v*/*v*). Diluted samples were analyzed in a UV-Vis spectrophotometer (Agilent Technologies cary series 100 UV-Vis) at 450 nm, using *n*-hexane as a reference, as described by Rougereau et al. [32]. The carotenoid content was quantified as mg β-carotene/kg algae (d.w.), according to the British Standard B.S. 684, Section 2.20:1977 [33]. Three replicates of each sample were analyzed.

#### 2.2.4. Total Phenolic Compounds and Total Flavonoids

The total phenolic content was determined according to the Folin–Ciocalteu method, as described by Matanjun et al. [34]. Methanolic extract of *P. canaliculata* (0.05 mL) was mixed with 0.05 mL of methanol, 1.6 mL of Milli Q water, and 0.1 mL Folin–Ciocalteu reagent. The mixture was kept in the dark for 5 min. Then, 0.3 mL of 20% (*m*/*v*) sodium carbonate solution was added, and finally, the mixture was again kept in the dark, at room temperature, for 1 h. After incubation, the absorbance of the solution was measured at wavelength 765 nm against methanol in a UV/VIS double beam spectrophotometer (Agilent Technologies cary series 100 UV-Vis). For the calibration curve, gallic acid was used in concentrations from 0.47 to 19.05 µg/mL. Samples were treated following the same procedure. The obtained standard curve had a determination coefficient, R^2^, of 0.994 (10 data points).

For the determination of total flavonoids, 0.25 mL of methanolic extract of *P. canaliculata* was used in which 0.25 mL of methanol, 2 mL dH_2_O, and 0.15 mL of NaNO_2_ 5% (*m*/*v*) were added. The mixture was kept in the dark for 5 min. Then, 0.15 mL of AlCl_3_ (10% *m*/*v*) was added, and the final mixture stayed in the dark for 6 min. Finally, 1 mL of NaOH aqueous solution (4% *m*/*v*) and 1.2 mL of dH_2_O were added, and the mixture was vortexed. The absorbances were determined at 510 nm, using 1 cm cell, in a UV/VIS double beam spectrophotometer (Agilent Technologies cary series 100 UV-Vis), with distilled water as blank [35]. Two calibration curves were made using quercetin (18 data points; R^2^ = 0.976) and catechin (10 data points; R^2^ = 0.999). Quercetin and catechin samples were treated following the same procedure. The results were expressed in milligram of catechin or quercetin equivalent/kg of algae (d.w.). Three replicates of each sample were analyzed.

#### 2.2.5. Determination of Antioxidant Activity

##### DPPH Radical Scavenging Assay

The antioxidant activity by the DPPH (2,2-diphenyl-1-picrylhydrazyl-hydrate) radical scavenging assay was determined as described by Brand-Williams et al. [36]. The reagent was prepared by dissolving 2.7 mg of DPPH in 100 mL methanol. Then, 3.9 mL of the freshly prepared DPPH reagent was mixed with 0.1 mL of methanolic extract of *P. canaliculata*. The mixture was vortexed and incubated in the dark, at room temperature, for 30 min. After incubation, the absorbance was read at 515 nm (A_sample_) against methanol using a UV/Vis spectrophotometer (Agilent Technologies cary series 100 UV-Vis). Furthermore, the absorbance at 515 nm for the DPPH reagent (3.9 mL) with 0.1 mL of methanol was determined, which corresponds to the maximum absorbance (A_max_) value in the calculations. The results were expressed as % radical scavenging activity (% RSA) that was calculated by the following equation:%RSA = 100 × (A_max_ − A_sample_)/A_max_(1)

For the calibration curve, the synthetic antioxidant Trolox (6-hydroxy-2,5,7,8-tetramethylchroman-2-carboxylic acid), an analog of vitamin E, was used. For the stock solution, Trolox (20 mg) was dissolved in methanol (100 mL). The stock solution was diluted in methanol in six different concentrations up to 5.25 µg/mL. The standard curve had a determination coefficient of 0.966, and the results obtained were expressed in the equivalent of Trolox (mg/kg of algae, d.w.). Three replicates of each sample were analyzed.

##### Ferric Reducing Antioxidant Power (FRAP)

The antioxidant capacity of the methanolic extract of *P. canaliculata* was estimated spectrophotometrically following the procedure of Benzie and Strain [37] with minor modifications. Ferric to ferrous ion reduction at low pH causes the formation of a colored ferrous-2,4,6-tri(2-pyridyl)-1,3,5-triazine (TPTZ) complex, which can be monitored at 595 nm. Freshly prepared FRAP reagent was used, which was obtained by mixing 25 mL of 0.3 M acetate buffer (pH 3.6), 2.5 mL of 10 mM TPTZ solution, and 2.5 mL of 20 mM ferric chloride solution. Immediately, 2.7 mL of FRAP reagent was mixed with 0.09 mL of the test solution and 0.27 mL distilled water. Samples were vortexed and incubated in the dark for 30 min at 37 °C. After incubation, the absorbance was read at 595 nm against the FRAP reagent.

For the calibration curve, the antioxidant Trolox was used. The stock solution (20 mg Trolox dissolved in 100 mL methanol) was diluted in methanol in four different concentrations (up to 6.4 µg/mL). Trolox samples were treated following the same procedure. The results obtained were expressed in the equivalent of Trolox (mg/kg of algae). Three replicates of each sample were analyzed.

### 2.3. Ultrasound-Assisted Extraction of Bioactive Compounds to the Sunflower Oil

Ultrasound-assisted extraction (UAE) of bioactive compounds from lyophilized *P. canaliculata* was performed directly into the oil using an ultrasound bath (Transsonic TS 540; ultrasound frequency of 35 kHz). The assays were performed according to an experimental central composite rotatable design (CCRD), which allows the use of the response surface methodology (RSM). The UAE time and the *P. canaliculata* concentration varied simultaneously with a minimum number of trials using RSM. This methodology has the advantage of being less expensive and time-consuming than the classical methods and provides the possibility to estimate interactions between factors [38]. In CCRD, five levels for each variable were applied. The use of five levels for each factor/variable allows the fitting of second-order polynomials to the experimental data points and, hence, to fit curved surfaces to the experimental data. The CCRD matrix, as a function of two factors (UAE time and algae load), consists of a group of four factorial points, a group of four star points and four central points (Table 1). The repetition of the central point enables an estimation of the variance of the experimental error, which is assumed to be constant along the experimental domain [39]. In addition to the experiments dictated by the CCRD, one extra sample (sample 13) was prepared and analyzed. Sample 13 consists of sunflower oil submitted for 20 min to ultrasonic bath and corresponds to the control sample (Table 1).

The lyophilized alga was added to 40 mL of sunflower oil in concentrations between 5% and 20% (*m*/*v*) (Table 1). The samples were vortexed and subsequently subjected to ultrasound-assisted extraction for 5 to 20 min (Table 1). Then, the samples were centrifuged (10 min at 3000 rpm) to separate the solid residue from the supplemented oil. Finally, the supplemented oils were stored, protected from light, in the refrigerator, at 4 °C, until further analyses.

### 2.4. Analyses of the Oil

#### 2.4.1. Chemical Quality Parameters of Supplemented Sunflower Oils

The acidity of the original and supplemented oil samples was determined by titration with 0.1 M potassium hydroxide solution, in accordance with the Commission Regulation (EEC) No.2568/91 [40] and was expressed in acid value (mg KOH/g oil). The presence of primary and secondary oxidation products in original and supplemented oil samples was determined in accordance with the Commission Regulation (EEC) No. 2568/91. Three replicates of each sample were analyzed.

#### 2.4.2. Sensory Analyses of Supplemented Sunflower Oils

The supplemented sunflower oil samples were submitted to a preliminary sensory analysis to evaluate the presence of off-odors. A group of 7 trained assessors on olive oil sensory analysis was used [40]. Coded samples were presented to the panelists in covered glasses at 28–30 °C. They were asked to smell the samples, detect, describe, and quantify the intensity of eventual off-flavors. A discontinuous and structured scale, from 1 (very slight intensity) to 5 (very strong intensity) was used. For all the samples, the mean and the median (as used for virgin olive oil sensory evaluation) of the detected off-flavors were calculated and used as the response of the panel.

#### 2.4.3. Chlorophyll Pigments and Carotenoids

The chlorophyll and carotenoid contents of the oil samples were determined as previously described (cf. 2.2.3) and expressed in mg of pheophytin a/kg oil and as β-carotene (mg/kg oil). Three replicates of each sample were analyzed.

#### 2.4.4. Determination of Chromatic Characteristics

The color of the samples was analyzed according to the Portuguese Standard NP 937 from 1972 [41]. The method and results are presented as Appendix A.

#### 2.4.5. Determination of Phenolic Compounds

##### Preparation of Phenolic Extracts

In order to obtain phenolic extracts, 20 g of oil samples were added to 40 mL of *n*-hexane and 40 mL of methanol-water solution (80:20, *v*/*v*) [42]. The mixture was vortexed for 5 min, and the lower methanolic phase was removed from the upper oil phase in a separatory funnel. The second washing of the oil phase was carried out by the addition of 20 mL of methanol-water solution (80:20; *v*/*v*). The mixture was vortexed for 5 min, after which methanolic phases were paper filtrated. Finally, the oil residues in the methanolic phase were removed by washing with 10 mL of *n*-hexane. The mixture was vortexed for 1 min and centrifuged for 10 min under 3500 rpm. After centrifugation, the upper layer (oil phase) was removed from the methanolic phase. The methanolic phase was evaporated until dry matter at 40 °C and 30 mbar [42]. Dry matter was stored at −18 °C until further analyses. Immediately before the subsequent analyses, these samples were dissolved in 2 mL of methanol. The quantification of total phenolic compounds and flavonoids and the determination of antioxidant activity were performed in triplicate with these extracts.

##### Determination of Total Phenolic Compounds and of Total Flavonoids

Total phenolic and total flavonoid contents were determined according to the methods described in Section 2.2.4. and expressed as mg of gallic acid equivalent/kg of supplemented sunflower oil and as mg of catechin or quercetin equivalent/kg of supplemented sunflower oil, respectively.

#### 2.4.6. Determination of Antioxidant Activity

Antioxidant activity in supplemented oil samples was determined by the DPPH radical scavenging assay (expressed as equivalent mg of Trolox/kg of supplemented sunflower oil) and by the ferric reducing antioxidant power (expressed as equivalent mg of Trolox/kg of supplemented sunflower oil), following the methods previously described (cf. Section 2.2.5).

### 2.5. Accelerated Oxidation Tests

The oil obtained by supplementation with 12.5% algae under the highest UAE time (20 min) was submitted to accelerated oxidation conditions. This oil was stored in 10 mL amber glass flasks (full of oil to avoid contact with oxygen) at 60 °C, in the dark, for 12 days. In parallel, non-supplemented sunflower oil was submitted to the same storage conditions. Aliquots corresponding to individual flasks were collected periodically and chemically analyzed.

### 2.6. Statistical Analysis

The obtained results were used to calculate the linear and quadratic effects of each factor, as well as the interaction (Alga concentration × UAE time) on each response. A response surface described by a second-order polynomial equation was fit to each set of results, using the software “Statistica”, version 7, from Statsoft, Tulsa, OK, USA. The significance of the effects of the factors was evaluated by ANOVA, where effects were considered significant when *p* ≤ 0.05. However, when the removal of a specific factor with *p* > 0.05 resulted in a decrease in the goodness of fit of the model, that factor was retained in the model, as suggested by Haaland [43]. The quality of the adjustment of the fitted models was evaluated by the determination coefficient (R^2^) and by the determination adjusted coefficient (R^2^_adj_) [39,43]. High values of both R^2^ and R^2^_adj_ suggest a good fit of the model to the experimental data. Values of R^2^ should be at least higher than 0.75 to have a good fit of the model to the experimental results. When above 0.90, it indicates a very good fit [43].

## 3. Results and Discussion

### 3.1. Pelvetia Canaliculata Characterization

#### 3.1.1. Proximate Composition

*P. canaliculata* was characterized concerning total protein, lipid, carbohydrates, and ash contents expressed as a percentage of dry weight. The algae biomass was mainly constituted by carbohydrates (65.76% ± 0.43%) followed by ash (21.40% ± 0.04%), protein (7.72% ± 0.13%), and lipids (5.12% ± 0.41%). These results are in agreement with previous studies regarding the characterization of *P. canaliculata* collected in Norway [44] and in the United Kingdom [45]. Nevertheless, in the study carried out by Badmus et al. [45], significantly lower amounts of protein (less than 2%) were reported, which may be justified by the use of the Bradford method that only quantifies soluble proteins [46].

#### 3.1.2. Chlorophyll and Carotenoid Pigment Contents

Chlorophyll pigments are responsible for the green color of many plants and algae. However, when in a food matrix, chlorophyll pigments may act as prooxidants [47]. Carotenoids are natural pigments responsible for yellow and orange colors and antioxidant activity.

The quantification of chlorophyll pigments (expressed as pheophytin a), carotenoids (expressed as β-carotene) in *P. canaliculata* was performed, and the results are shown in Table 2. The analysis showed the presence of 602 mg of pheophytin a, and 236 mg of β-carotene per kg of lyophilized *P. canaliculata*. These results were used to calculate the extraction yield of each group of pigments from the algae to the sunflower oil in each experiment.

Hupel et al. [48] determined chlorophyll and carotenoid contents in *Pelvetia canaliculata* harvested in France. The authors quantified 480 mg of chlorophyll a/kg of dried extract, 130 mg of chlorophyll c/kg of dried extract, and 120 mg of β-carotene/kg of dried extract. Thus, the total amount of chlorophylls (a and c) determined by the authors was 610 mg of chlorophyll/kg of the dried extract, which is similar to the value obtained in the present study. The carotenoid content determined by Hupel et al. [48] was about twice smaller the value determined in the present study, which can be explained by the use of different methods of extraction and quantification, and by the different environmental conditions that the algae were exposed to.

#### 3.1.3. Determination of Total Phenolic Compounds, Flavonoids, and Antioxidant Activity

Phenolic compounds are presented by a diverse group of secondary metabolites that abundantly occurs in algae and plants. In addition to their antioxidant activity, they also have a synergistic effect with other antioxidant molecules such as ascorbic acid, β-carotene, and α-tocopherol [49]. Flavonoids are important natural pigments with antioxidant activity. Thus, both carotenoids and flavonoids are attractive compounds for the food industry.

The total phenolic content (expressed in gallic acid), the flavonoids (expressed as quercetin or catechin), and the antioxidant activity, assayed by the DPPH radical scavenging method and by the ferric reducing antioxidant power (FRAP), of *P. canaliculata* methanolic extracts, are shown in Table 3. The alga presented 5 544 mg of gallic acid/kg of lyophilized algae and flavonoid content of 2966 mg of catechin/kg of lyophilized algae (7849 mg of quercetin/kg). Concerning the antioxidant activity, *P. canaliculata* showed a radical scavenging activity of 86% (895 mg of Trolox/kg of lyophilized algae) and antioxidant activity, determined by FRAP assay, of 3592 mg of Trolox/kg of lyophilized algae.

Hupel et al. [48] determined the total phenolic content of *P. canaliculata* extracts, and the results varied between 5.82 and 8.92 mg of gallic acid/g dried extract, depending on the radiation that the algae were exposed. The phenolic content obtained in the present study (5.544 mg gallic acid/g lyophilized algae) was similar to the lower amount obtained by Hupel et al. [48]. Again, the slight differences may be due to the environmental conditions to which the algae were exposed and to the extraction methods used.

### 3.2. Quality Parameters of the Supplemented Sunflower Oil

The ultrasound-assisted supplementation of sunflower oil with *P. canaliculata* compounds was conducted according to a CCRD as a function of ultrasound-assisted extraction time and algae concentration (Table 1). The obtained results were used to calculate the linear and quadratic effects of each factor, as well as their interaction effect, on each response and to fit a response surface described by a polynomial equation to each set of results.

#### 3.2.1. Acidity

The acidity of an oil is a quality parameter related to the extent of hydrolysis of acylglycerols responsible for the release of free fatty acids. In the present study, the acidity assayed as acidity value (mg KOH/g oil) varied from 0.08 to 0.16 mg KOH/g oil (Table 4). These values are lower than the maximum value allowed for refined vegetable oils (0.6 mg KOH/g oil) [3]. The statistical analysis of the results showed that there are no significant effects of treatment conditions on the acidity of supplemented oils. In addition, no significant differences between supplemented and non-supplemented oil samples were found. This means that extraction time and the amount of algae did not affect the acidity of the sunflower oil.

#### 3.2.2. Oxidation Products

The oxidation state of the oils was evaluated by absorbance measurements at 232 and 270 nm, K_232_ and K_270_. Primary oxidation products (conjugated dienes, namely conjugated hydroperoxides) absorb at 232 nm, while secondary oxidation products (e.g., unsaturated ketones, short-chain fatty acids, and aldehydes) absorb at 270 nm. However, conjugated trienes formed during oil refining, in the bleaching process by active earths, also absorb at 270 nm. Thus, in principle, K_270_ should not be applicable to refined oils [50]. In fact, there are no legal limits for K_232_ and K_270_ for refined oils. However, in this study, K_232_ and K_270_ values were used to assess the oxidation state of the samples after ultrasound treatment since all the values were compared to those of the original sunflower refined oil.

The statistical analysis of the results showed that there were no significant effects of treatment conditions on the oxidation of supplemented oil samples. This means that the extraction time and algae concentration had no impact on oil oxidation.

#### 3.2.3. Sensory Analysis

Consumer perception and acceptance of a food are particularly based on its sensory properties. All samples of supplemented oil presented sensory defects, described as the smell of algae/fish/rotten fish, with average sensory scores between 1.3 and 3.3 and median values between 1 (very slight intensity) and 3 (medium intensity) in a discontinuous scale from 1 to 5 (Table 4). The intensity of defects showed to slightly increase with the ultrasound-assisted extraction time. However, the statistical analysis of the results showed that neither algae concentration nor UAE time had any significant effects on the intensity of sensory off-flavors in the supplemented samples. It seems that the volatile compounds are transferred from the algae to the oil very rapidly. The lack of relationship between the algae concentration and the sensory defects may be ascribed to the low solubility of these volatiles in the oil.

### 3.3. Chlorophyll Pigments and Carotenoids Contents in Supplemented Sunflower Oil

Table 5 shows the amount of chlorophyll pigments and carotenoids present in supplemented sunflower oil samples, as well as the extraction yield (*Y*) for each group of compounds extracted from the algae to the oil during UAE. Extraction yields were calculated as follows:(2)$$Y=\frac{\left[C_{so}-C_{o}\right]}{C_{\mathrm{algae}}}\times 100$$
where $C_{so}$ is the concentration of the compound in the supplemented oil, $C_{o}$ is the initial concentration of that compound in the original oil submitted to 20 min ultrasounds, and $C_{\mathrm{algae}}$ is the amount of that compound available from the algae in contact with the oil.

#### 3.3.1. Chlorophyll Pigments

The presence of chlorophyll pigments in *P. canaliculata* (602 mg pheophytin a/kg of *Pelvetia*) suggests that the addition of the algae to the oil will cause an increase in their content in the oil (Table 5). The final content of chlorophylls in the supplemented oils varied from 25.84 to 72.11 mg pheophytin a/kg oil. These values represent 54.7% to 88.4% of the green pigments of the algae that were extracted from the oil during the UAE.

The statistical analysis of the results showed that the content of chlorophyll pigments increased linearly with algae concentration (*p* = 0.00002) and decreased with ultrasound-assisted treatment time (*p* = 0.021). In addition, a significant quadratic negative effect of algae concentration (*p* = 0.005) was observed, showing that the extraction of chlorophyll pigments from *Pelvetia* to the oil can be described by a convex surface as a function of algae content. No significant effects of the quadratic effect of time (*p* = 0.54) and of the interaction of the two factors (*p* = 0.15) were found regarding chlorophyll extraction to the oil. Therefore, the amount of chlorophylls in the oil can be fitted to the response surface shown in Figure 1a and described by the following second-order polynomial equation:[pheophytin a] (mg/kg oil) = −0.775 + 8.128 × [algae] − 0.216 × [algae]^2^ − 0.711*t*(3)
where [algae] is the concentration of the algae (d.w.), expressed in % (*m*/*v*), and t corresponds to ultrasound extraction time (min). This model shows a remarkable high fit to the experimental points with a determination coefficient R^2^ equal to 0.951 and an adjusted determination coefficient (R^2^_Adj_) of 0.932.

#### 3.3.2. Carotenoids

Carotenoids appear naturally in oils. However, most of them are lost during the bleaching operation in oil refining. Table 5 shows the final amounts of carotenoids in the supplemented oils, expressed in β-carotene, with values varying from 3.87 to 8.23 mg/kg oil. Only 10.32% to 28.08% of the algal carotenoids migrated to the oils. Statistical analysis of the results showed that carotenoid ultrasound-assisted extraction from the algae to the oil linearly increased with algae concentration (*p* = 0.009). Moreover, extraction time had a significant quadratic positive effect (*p* = 0.028) on carotenoids extraction. No significant contribution of quadratic effect of algae concentration, linear effect of time, or the interaction (UAE time x algae concentration) was found. Algae concentration showed a stronger impact on the amount of carotenoids extracted to the oil as compared to ultrasound extraction time. Therefore, the experimental results could be fitted to the response surface in Figure 1b (R^2^ = 0.771; R^2^_Adj_ = 0.720) described by Equation (5):(4)$$\left[\beta -\mathrm{carotene}\right]\left(\mathrm{mg}/\mathrm{kg oil}\right)=3.669+0.202\times \left[\mathrm{algae}\right]+0.0007\;t^{2}$$
where [algae] is the concentration of the algae (d.w.), expressed in % (*m*/*v*), and *t* corresponds to ultrasound extraction time (min).

These results are similar to those obtained by Goula et al. [27] in UAE from pomegranate wastes to sunflower oil. They found that the increase in the ratio of pomegranate waste: sunflower oil increased the carotenoid content in the supplemented oil. Goula et al. [27] also observed that an increase in the ultrasound extraction time, from 10 to 30 min, caused an increase in the amount of carotenoids extracted from pomegranate wastes into the vegetable oils.

### 3.4. Color Evaluation

The color is a sensory parameter that depends on the presence and amount of pigments, such as chlorophylls, carotenoids, flavonoids, and anthocyanins. This parameter can strongly influence the acceptance of a product by the consumer. In this context, the color of the samples of supplemented sunflower oil was determined and compared to the color of non-supplemented sunflower oil. Results regarding color evaluation by the chromatic characteristics are shown in Appendix A. The dominant wavelength of supplemented samples was constant and equal to 577 nm, which shows to be independent of the algae concentration or the UAE time used in each experiment. Color purity of supplemented samples could be described by a second-order polynomial as a function of both UAE time and algae concentration (Appendix A). Moreover, color purity showed to increase with chlorophyll and carotenoid contents in the supplemented oils, following logarithmic models (Appendix A). All supplemented samples showed a yellow/green color, as a result of pigment extraction from the algae to the refined colorless sunflower oil. The color of supplemented oils can be perceived as a positive attribute. As an example, Figure 2 shows the color of the original oil and the color of the supplemented oil sample obtained by 20 min UAE using 12.5% *m*/*v*
*Pelvetia*.

### 3.5. Phenolic Compounds and Antioxidant Activity

#### 3.5.1. Phenolic Compounds

The content of total phenolic compounds in supplemented sunflower oil samples, as well as the extraction yield, are presented in Table 6. *Pelvetia* methanolic extract had 5544 mg equivalent of gallic acid/kg lyophilized alga. So, it was expected that the addition of this alga to the oil would increase the phenolic content. In fact, all the supplemented oil samples presented higher phenolic contents (8.68 to 26.41 equivalent mg of gallic acid/kg oil) than the original refined sunflower oil (0.4 equivalent mg of gallic acid/kg oil). However, the extraction yields from the algae to the oils were rather low, ranging from 0.92% to 3.10% (Table 6). This may be explained by the low solubility of the phenolic compounds of *Pelvetia* in the oil.

The statistical analysis of the CCRD results showed that algae concentration and UAE time had a significant negative linear effect (*p* = 0.0015 and *p* = 0.001, respectively) on phenolic extraction from algae to the oil. On the other hand, the interaction of these two factors (algae concentration × time) had a significant positive linear effect (*p* = 0.005) on the extraction of these compounds to the oil. The quadratic positive effects of extraction time (*p* = 0.011) and *Pelvetia* concentration indicate that the phenolic content of supplemented samples can be fitted to a concave response surface (Figure 3a) as a function of both factors, described by the following second-order polynomial equation (R^2^ = 0.961; R^2^_Adj_ = 0.922):(5)$$\begin{array}{cc} \left[\mathrm{gallic}\;\mathrm{acid}\right]\left(\mathrm{mg}/\mathrm{kg}\mathrm{oil}\right)= & 36.02-1.98\left[\mathrm{algae}\right]+0.04{\left[\mathrm{algae}\right]}^{2}-3.51t\;+\\ & 0.10t^{2}+0.14\left[\mathrm{algae}\right]t \end{array}$$
where [algae] is the concentration of the algae, expressed in % (*m*/*v*), and *t* corresponds to ultrasound extraction time (min).

Jiménez-Escrig et al. [51] evaluated the phenolic content of aqueous and organic extracts of brown and red algae, and they concluded that acetone:water solution (70:30, *v*/*v*) was more efficient in the extraction of phenolic compounds. In our study, the highest amount extracted corresponds to 3.1% of the phenolic compounds from the algae. Conversely, López et al. [52], who determined the phenolic content of different extracts of the brown alga *Stypocaulon scoparium*, concluded that the aqueous extract had the highest phenolic content.

#### 3.5.2. Flavonoids

In this study, the flavonoid content in the algae and supplemented sunflower oil samples was determined and expressed as equivalent of catechin and equivalent of quercetin per kg of oil (Table 6). Quercetin and catechin are the flavonoids that are the most widely distributed in edible plants. Moreover, several authors, such as Saad et al. [53], refer to the presence of catechin and quercetin in algae (Red Marine Alga *Alsidium corallinum*). In this work, the authors expressed flavonoids using more than one standard to make a comparison to other authors easier, as several are used in the literature (including rutin, beyond quercetin and catechin).

The algae present 2966 mg of catechin or 7849 mg of quercetin/kg lyophilized algae. Therefore, it was expected that supplemented oil had much more flavonoids than non-supplemented oil. However, the results presented in Table 6 show a limited extraction of flavonoids to the oil, with extraction yields varying from 0.85% to around 9.40%. These results can be explained by the low solubility of flavonoids of *Pelvetia* in the oil. Moreover, when the statistical analysis was performed on the CCRD results, no significant effects of algae concentration or UAE time on flavonoids extraction were observed.

Our results are in contrast with those obtained by Murugan and Iyer [54] and by Chan et al. [9]. Murugan and Iyer [54] determined the flavonoid content of methanol, chloroform, ethyl acetate, and aqueous extracts of the brown alga *Turbinaria ornata* and detected the highest flavonoid content (88.4 µg catechin equivalent/mL extract) in the chloroform extract, which was the less polar solvent tested. Chan et al. [9] quantified the flavonoid content of various *Gracilaria changii* extracts and concluded that the decrease in solvent polarity promoted higher flavonoid contents in the extracts. These results indicate that the flavonoids present in *Turbinaria ornata* and *G. changii* are mainly non-polar. The difference between our results and the results obtained by other authors may be explained by the fact that the solubility of flavonoids is affected by the nature of the solvent and the flavonoid structure [55]. Thus, we may conclude that the flavonoids present in *P. canaliculata* are mainly nonsoluble in oil.

#### 3.5.3. Antioxidant Activity

##### DPPH Assay

The results from the radical scavenging activity of the supplemented oil samples are shown in Table 7. Methanolic extract of *Pelvetia*, in a concentration of 6.28% *m*/*v*, has 86% of RSA and 895 equivalents of mg Trolox/kg lyophilized algae (Table 3). Therefore, as expected, the addition of the algae to the oil increased its radical scavenging activity. Nevertheless, the application of response surface methodology showed that there were no significant effects of algae concentration and/or ultrasound-assisted extraction time on radical scavenging activity of the supplemented oils.

Chan et al. [9] found a correlation between the DPPH radical scavenging activity and phenolic and flavonoid contents of various *G. changii* extracts. However, no relationship between phenolic or flavonoid contents and radical scavenging activity was detected in our study. Similarly, Cox et al. [56], who determined the free radical scavenging activity and phenolic content of various macroalgae (brown, red, and green), concluded that there was no direct relationship between phenolic content and free radical scavenging activity of macroalgae extracts.

##### FRAP Assays

The results of FRAP assays of the initial and supplemented oil samples are also presented in Table 7. Methanolic extract of *P. canaliculata* had 3592 mg ± 180 Trolox/kg algae (Table 3). Therefore, if the compounds responsible for antioxidant activity were liposoluble, an increase in the antioxidant activity of supplemented oils was expected. In fact, the samples of supplemented sunflower oil have higher antioxidant activity than the non-supplemented oil (Table 7). These results indicate that the compounds with antioxidant activity that migrated from the algae to the oil are mainly non-polar. Similarly, Chan et al. [9] studied the antioxidant activity, by the FRAP assay, of several *G. changii* extracts, using different solvents. They observed that the decrease in solvent polarity corresponded to an increase in the antioxidant activity of the extracts, which means that compounds that were responsible for antioxidant activity were mainly non-polar.

Statistical analysis of the obtained results showed that algae concentration and ultrasound extraction time have significant linear negative effects (*p* = 0.05 and *p* = 0.008, respectively) on the antioxidant activity of the samples. Ultrasound extraction time had a higher impact on the antioxidant activity of the compounds extracted to the oil than algae concentration. The quadratic positive effect of time and the linear interaction between the two factors are important enough to be considered in the response surface model.

Therefore, a response surface (Figure 3b), described by the following equation, was fitted to the experimental results (R^2^ = 0.863; R^2^_Adj_ = 0.771):(6)$$\left[\mathrm{Trolox}\right]\;\left(\mathrm{mg}/\mathrm{kg}\;\mathrm{oil}\right)=22.119-0.623\left[\mathrm{algae}\right]-1.958t+0.063t^{2}+0.078\left[\mathrm{algae}\right]t$$
where [algae] is the concentration of the algae (d.w.), expressed in % (*m*/*v*), and *t* corresponds to ultrasound extraction time (min).

Figure 3 shows that response surfaces fitted to phenolic content and to the antioxidant activity assayed by the FRAP method have a similar shape. In fact, a linear relationship, with a determination coefficient of 0.913, was found between these two parameters (Trolox = 0.648 [Phenols] + 4.409).

Chan et al. [9] also found a correlation between the phenolic content of the extracts and the antioxidant activity. However, they also found a relationship between flavonoid content and antioxidant activity and between radical scavenging activity and antioxidant activity, which was not observed in this study.

Our results showed that the antioxidant activity of the supplemented oils, associated with the presence of phenolic compounds, is not derived from radical elimination by hydrogen donation. Thus, the antioxidant activity of supplemented sunflower oil might be explained by other mechanisms, such as: (i) capture of species that initiate peroxidation; (ii) chelation of metal ions, preventing them from generating reactive species and decomposing lipid peroxides; (iii) chelation of singlet oxygen, preventing the formation of peroxides; or (iv) reduction in local oxygen concentrations [57,58,59].

Data from the literature are controversial about the relationship between phenolic content and antioxidant activity because the antioxidant activity depends on the structure of phenolic compounds, their polymerization degree, and on the solvents used for extraction of active compounds [51,52,60].

### 3.6. Accelerated Oxidation Test

From CCRD results, the oil sample with the highest phenolic content and antioxidant activity (DPPH and FRAP values) was obtained using 12.5% (*m*/*v*) of *Pelvetia* and submitted to 20 min UAE (Table 6 and Table 7; experiment 8). For the accelerated oxidation assay, a different batch of *P. canaliculata* was used to prepare supplemented oils under the same conditions as experiment 8.

Samples were stored in the dark at 60 °C for 12 days. The assessment of oxidative stability was performed by DPPH and FRAP assays on the samples along the experiment (Figure 4). Concerning radical scavenging activity, the initial sample used in this experiment had 46.05 mg ± 2.30 of Trolox/kg of oil while the sample nº 8 from CCRD (Table 7), from a different *Pelvetia* batch, had 14.99 mg ± 1.20 Trolox/kg of oil. The natural variability in *Pelvetia canaliculata* might be the cause of different results.

The supplemented sunflower oil had 10 times higher radical scavenging activity than the non-supplemented sample (4.42 mg ± 0.22 of Trolox/kg of oil). Along the experiment, radical scavenging activity of supplemented and non-supplemented oils remained constant, with small oscillations (Figure 4a). After 12 days at 60 °C, the radical scavenging activity of supplemented sunflower oil was 48.51 mg ± 2.43 of Trolox/kg of oil, while in the non-supplemented sample, it decreased to 2.97 mg ± 0.15 of Trolox/kg of oil. These results indicate that *Pelvetia* improved radical scavenging activity and, consequently, increased the oxidative stability of sunflower oil.

However, our results are not in accordance with the ones obtained by other authors. Yang et al. [61] studied the radical scavenging activity of soybean, rice, and cotton oils, kept at 62 °C for 24 days. Siraj et al. [62] studied the radical scavenging activity of canola and sunflower oils supplemented with pomegranate seed oil, kept at 62 °C for 60 days. In both studies, the authors observed a decrease in radical scavenging activity during the storage. This might indicate that the DPPH test is not adequate to follow the oxidative stability of sunflower oil supplemented with *Pelvetia* extracts under accelerated storage conditions.

Therefore, the antioxidant activity of the samples was also determined by the FRAP test (Figure 4b). On day 0, the supplemented sunflower oil had an antioxidant activity of 21.4 mg ± 1.1 of Trolox/kg that remained almost constant during 48 h under heated storage conditions. Then, the antioxidant activity of the supplemented sunflower oil decreased with storage time and reached 8.6 mg ± 0.4 of Trolox/kg of oil after 12 days. With respect to the non-supplemented oil, its initial antioxidant activity of 3.6 mg ± 0.2 of Trolox/kg of oil linearly decreased to 1.6 mg ± 0.1 of Trolox/kg of oil along 12 days at 60 °C. These results indicate that the supplementation of sunflower oil with *Pelvetia* extracts significantly improved its antioxidant activity. Our results are in accordance with Tinello et al. [63], who observed a decrease in the antioxidant activity of soybean oil enriched with turmeric and ginger over storage time (62 °C/28 days). In conclusion, FRAP seems to be a more accurate test than DPPH to follow the oxidative stability of sunflower oil supplemented with *Pelvetia* extracts and submitted to accelerated storage conditions (60 °C).

## 4. Conclusions

In this study, bioactive compounds were successfully extracted from the lyophilized alga *Pelvetia canaliculata* directly to the sunflower oil, using non-pollutant ultrasound-assisted extraction. The optimization of UAE conditions was performed by response surface methodology following a central composite rotatable design as a function of algae concentration and ultrasound-assisted extraction time. The extraction of compounds from *Pelvetia* did not affect the chemical quality parameters of the supplemented oils. Both chlorophyll and carotenoid contents increased after the supplementation, giving a yellow/green color to the oils. However, the addition of *Pelvetia* induced odor defects in the oil, from very slight to medium intensity, which may compromise consumers’ acceptability. Therefore, it would be necessary to deodorize the supplemented oils, to remove these off-flavors.

Moreover, phenolic content and antioxidant activity increased in all supplemented samples. A linear correlation between antioxidant activity variation (FRAP) and phenolic content was observed. The highest antioxidant activity was observed in the sample prepared with 12.5% *m*/*v* of *Pelvetia* and 20 min of UAE. Therefore, this sample was subjected to an accelerated oxidation test, performed in the dark at 60 °C for 12 days. FRAP assay showed to be the most adequate to follow the antioxidant activity of the supplemented and non-supplemented oils along accelerated oxidation test. Along this experiment, the antioxidant activity of supplemented oil was about six times the value of non-supplemented oil. Final samples showed 40% of initial antioxidant activity.

The results obtained in this study highlight the applicability of ultrasound-assisted extraction and the potential use of algae as a source of bioactive compounds in oils. This supplementation increases the oxidative stability, the biological and economical values of the sunflower oil.

## Figures and Tables

**Figure 1 foods-10-01732-f001:**
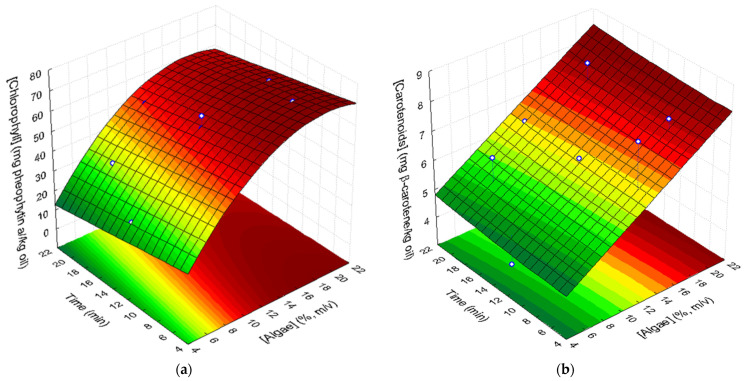
Response surfaces describing the effect of algae concentration and UAE time on (**a**) chlorophyll pigments (expressed in mg of pheophytin a/kg oil) and (**b**) carotenoids (mg β-carotene/kg oil) concentration in supplemented sunflower oil samples.

**Figure 2 foods-10-01732-f002:**
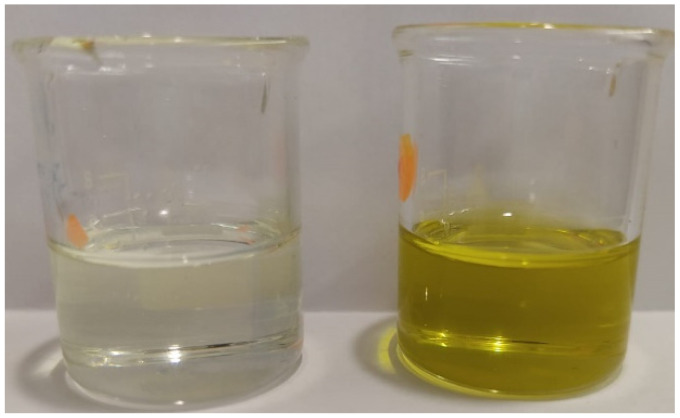
Non-supplemented sunflower oil (**left**) and supplemented sunflower oil (12.5% *m*/*v*
*Pelvetia*; 20 min UAE) (**right**).

**Figure 3 foods-10-01732-f003:**
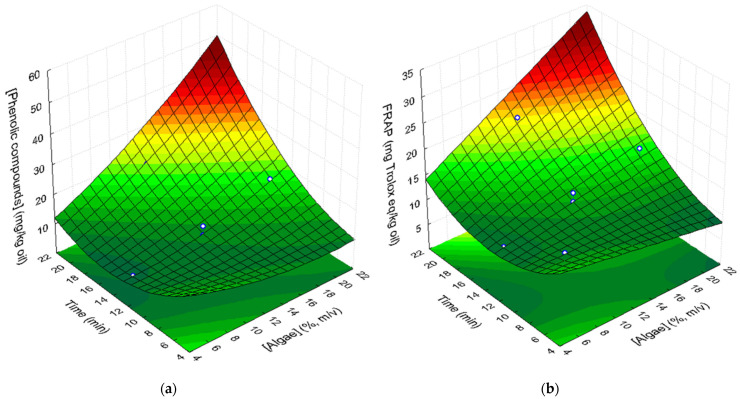
Response surfaces describing the effect of algae concentration and UAE time on (**a**) phenolic compounds concentration, expressed in equivalent mg of gallic acid/kg oil, and (**b**) antioxidant activity evaluated by the FRAP method in supplemented sunflower oil samples.

**Figure 4 foods-10-01732-f004:**
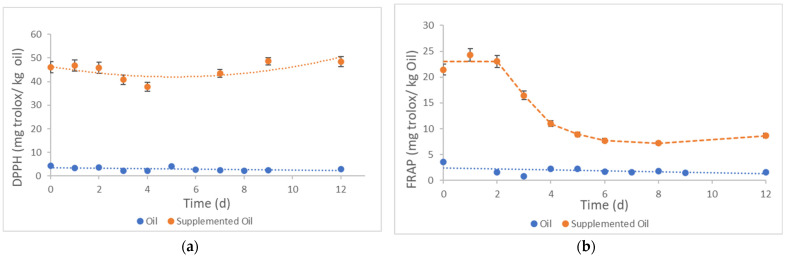
Accelerated oxidation tests carried out under dark at 60 °C, for 12 days, of non-supplemented and supplemented sunflower oil obtained by UAE with *Pelvetia* (12.5%, *m*/*v*) for 20 min. The antioxidant activity was assayed by (**a**) DPPH and (**b**) FRAP methods.

**Table 1 foods-10-01732-t001:** CCRD and blank experiments followed for ultrasound-assisted extraction (UAE) of bioactive compounds from *Pelvetia canaliculata* to sunflower oil, as a function of UAE time treatment and algae concentration.

	Assay	[Algae]	UAE Time	[Algae] (% *m*/*v*) Decoded Values	UAE Time (min) Decoded Values
		Coded Value	Coded Value		
Factorial points	1	−1	−1	7.2	7.2
	2	−1	1	7.2	17.8
	3	1	−1	17.8	7.2
	4	1	1	17.8	17.8
Star points	5	$-\sqrt{2}$	0	5.0	12.5
	6	$\sqrt{2}$	0	20.0	12.5
	7	0	$-\sqrt{2}$	12.5	5.0
	8	0	$\sqrt{2}$	12.5	20.0
Central points	9	0	0	12.5	12.5
	10	0	0	12.5	12.5
	11	0	0	12.5	12.5
	12	0	0	12.5	12.5
Blank	13	_	_	0	20.0

**Table 2 foods-10-01732-t002:** Results for the concentration of chlorophyll and carotenoid pigments in *P. canaliculata* (d.w.); results are the average of three repetitions; STD = standard deviation.

Pigments	Amount (mg/kg Algae)	STD
Chlorophyll [pheophytin a]		
	602	30
Carotenoids [β-carotene]		
	236	12

**Table 3 foods-10-01732-t003:** Concentration of phenolic compounds and antioxidant activity of *P. canaliculata* (d.w.); results are the average of three repetitions; STD = standard deviation.

Determination	Amount (mg/kg Algae)	STD
Total phenolic content [gallic acid]	5544	277
Flavonoids [quercetin]	7849	393
Flavonoids [catechin]	2966	148
DPPH, % (RSA)	86	5
DPPH, [Trolox]	895	52
FRAP, [Trolox]	3592	180

**Table 4 foods-10-01732-t004:** CCRD results for acid value, oxidation products, and sensory analysis for each sample of supplemented sunflower oil. Sample 13 is the original refined sunflower oil submitted to a 20 min ultrasound treatment (average values and standard deviations, STD, of three replicates).

Assay	Acid Value (mg KOH/g Oil)		Oxidation Products						Sensory Analysis		
		STD	K_232_	STD	K_270_	STD	ΔK	STD	Average	STD	Median
1	0.12	0.02 × 10^−1^	3.16	0.16	1.21	0.06	0.12	0.02	1.3	0.5	1.0
2	0.12	0.01 × 10^−1^	3.25	0.16	1.26	0.06	0.16	0.01	2.1	0.9	2.0
3	0.12	0.01 × 10^−1^	3.24	0.16	1.25	0.06	0.16	0.02	2.2	0.9	2.0
4	0.12	0.03 × 10^−2^	3.24	0.16	1.21	0.06	0.13	0.01	2.1	1.5	2.0
5	0.12	0.01 × 10^−1^	3.17	0.16	1.21	0.06	0.10	0.01	1.8	1.2	1.5
6	0.12	0.03 × 10^−1^	3.25	0.16	1.21	0.06	0.13	0.02	3.3	1.4	3.0
7	0.16	0.04	3.33	0.17	1.26	0.06	0.17	0.02	1.6	0.5	2.0
8	0.08	0.04	3.30	0.16	1.24	0.06	0.15	0.01	3.0	1.5	3.0
9	0.12	0.01 × 10^−1^	3.33	0.17	1.27	0.06	0.15	0.02	2.3	1.4	2.0
10	0.12	0.05 × 10^−1^	3.28	0.16	1.21	0.06	0.13	0.01	2.8	1.6	3.0
11	0.12	0.02 × 10^−1^	3.24	0.16	1.22	0.06	0.14	0.02	2.8	1.0	2.0
12	0.12	0.03 × 10^−1^	3.42	0.17	1.31	0.07	0.32	0.02	2.2	1.1	2.0
13	0.12	0.02 × 10^−2^	3.18	0.16	1.28	0.06	0.10	0.01	___		___

**Table 5 foods-10-01732-t005:** CCRD results for the concentration of chlorophyll pigments and carotenoids for each sample of supplemented sunflower oil. Sample 13 is the original refined sunflower oil with a 20 min ultrasound treatment; (average values and standard deviations, STD, of three replicates).

Assay	Chlorophyll Pigments			Carotenoids		
	(mg Pheophytin a/kg Oil)	STD	Extraction Yield (%)	[β-Carotene] (mg/kg Oil)	STD	Extraction Yield (%)
1	37.15	1.86	78.00	4.87	0.04	20.20
2	37.90	1.89	79.56	6.34	0.03	28.08
3	72.11	3.61	61.24	7.93	0.03	14.81
4	61.70	3.08	52.40	8.20	0.04	15.39
5	25.84	1.29	78.05	3.87	0.06	21.39
6	66.15	3.31	50.01	6.45	0.06	10.32
7	66.75	3.34	80.72	8.23	0.06	21.99
8	52.27	2.61	63.21	6.55	0.03	16.84
9	56.27	2.81	68.04	5.86	0.04	14.70
10	53.17	2.66	64.30	5.90	0.05	14.80
11	57.26	2.86	69.24	6.46	0.04	16.55
12	63.34	3.17	76.60	5.08	0.05	12.29
13	0.22	0.01	-	1.10	0.03	-

**Table 6 foods-10-01732-t006:** CCRD results for the concentration of total phenolic compounds and flavonoids of each sample of supplemented sunflower oil. Sample 13 is the original refined sunflower oil with a 20 min ultrasound treatment (average values and standard deviations, STD, of three replicates; * the phenolic compounds of sample 9, corresponding to one repetition of the central point, could not be determined due to sample shortage).

Assay	Phenolic Content			Flavonoids					
	[Gallic Acid] (mg/kg Oil)	STD	Extraction Yield (%)	[Quercetin] (mg/kg Oil)	STD	Extraction Yield (%)	[Catechin] (mg/kg Oil)	STD	Extraction Yield (%)
1	10.26	0.51	2.47	84.23	4.21	4.60	31.76	1.59	4.65
2	11.09	0.55	2.68	110.69	5.53	9.28	41.89	2.09	9.40
3	9.47	0.47	0.92	94.24	4.71	2.58	35.59	1.78	2.61
4	26.41	1.32	2.64	97.77	4.89	2.83	36.95	1.85	2.86
5	7.79	0.39	2.67	73.24	3.66	3.82	27.56	1.38	3.,87
6	18.71	0.94	1.65	93.57	4.68	2.25	35.34	1.77	2.28
7	11.67	0.58	1.63	85.01	4.25	2.73	32.07	1.60	2.76
8	21.91	1.10	3.10	77.25	3.86	1.94	29.09	1.45	1.96
9	*	-	*	*	-	*	*		*
10	10.45	0.52	1.45	90.9	4.55	3.33	34.32	1.72	3.37
11	8.68	0.43	1.19	84.20	4.21	2.64	31.76	1.63	2.68
12	13.49	0.67	1.89	66.58	3.33	0.85	25.01	1.25	0.86
13	0.40	0.02	-	58.26	2.91	-	21.83	1.09.	-

**Table 7 foods-10-01732-t007:** Antioxidant activity of each sample of supplemented sunflower oils obtained under the conditions dictated by the CCRD followed. Sample 13 is the original refined sunflower oil with a 20 min ultrasound treatment (average values and standard deviations, STD, of three replicates; * FRAP assays of sample 9, corresponding to one repetition of the central point, could not be determined due to sample shortage).

Assay	DPPH				FRAP	
	% RSA	STD	Eq. Trolox (mg/kg Oil)	STD	Eq. Trolox (mg/kg Oil)	STD
1	63.8	3.2	12.34	0.62	11.85	0.59
2	47.4	2.4	8.96	0.45	12.36	0.62
3	49.8	2.5	9.59	0.48	9.86	0.49
4	60.0	3.0	11.50	0.58	19.15	0.96
5	48.7	2.4	9.33	0.47	9.35	0.47
6	74.4	3.7	14.47	0.72	16.58	0.83
7	57.6	2.9	10.95	0.55	10.40	0.52
8	77.5	3.9	14.99	0.75	21.44	1.07
9	51.7	2.6	9.92	0.50	*	-
10	51.1	2.6	9.78	0.49	8.91	0.45
11	49.5	2.5	9.41	0.47	11.99	0.60
12	55.0	2.8	10.60	0.53	13.74	0.69
13	7.5	0.3	1.10	0.06	3.61	0.18

## Data Availability

Data will be available on request to the authors.

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
