# Peer review of "Optimization of Ultrasound-Assisted Extraction of Bioactive Compounds from Pelvetia canaliculata to Sunflower Oil"

_foods, 2021, doi:10.3390/foods10081732_

Round 1
Reviewer 1 Report
See attached file

Author Response
Reviewer comments and our replies
We thank the Reviewer #1 for the recognition of the value of our study, and for the time and effort that was obviously put in here to improve our manuscript.
We believe we have integrated all the comments of Reviewer #1 listed below into this revision.
Reviewer 1
The objective of this study was to formulate a refined sunflower oil modified by direct supplementation with bioactive compounds from Pelvetia canaliculata by ultrasound assisted extraction. The work analyses the effect of algae concentration and the extraction time on pigments, total phenols and flavonoids and antioxidant activity measured by DPPH and FRAP test. An accelerated oxidation study and a small sensory study are also carried out. This is not a very innovative study. It is evident that the addition of an algae extract to a refined oil increases the pigment and phenol content and also provides antioxidants. In addition, as expected, the fortified oil has unpleasant flavours that may compromise its acceptability. Here are some comments that could contribute to improve the manuscript and recommendations for further studies
Concerning the lack of innovation, we would like to inform you that the novelty of this work (alga and UAE used) was recognized by the grant of a Portuguese Patent, where some of the co-authors of this paper are the inventors (nº 115610; priority date: 27 June 2019), that was published by the Portuguese Institute of Intellectual Property (INPI) on the 25th June 2021.
- Supplementation: How many replicates of each treatment have been done?
Our reply: The great advantage of using response surface experimental designs is the reduction of the number of experiments and, at the same time, the possibility of evaluating eventual interactions between factors (variables), which cannot be evaluated by a classical one-variable-at-a-time approach (OVAT). In the experimental design followed (Central Composite Rotatable Design), each factor (US time and Alga concentration) was tested at 5 different levels (values). Only the central point is repeated, and the variance observed in the repetition of this point is assumed to be constant over the entire experimental region. This explanation was already in the first version of the manuscript (lines 220-225).
All the chemical analysis (alga and oil samples) were carried out in triplicate. This information was added in the materials and methods section at the end of the description of each analytical procedure.
Please, specify the ultrasound frequency.
Our reply: The ultrasound frequency of the equipment is 35 kHz. This information was added to the section “2.3. Ultrasound-assisted extraction of bioactive compounds to the edible oil”.
- The sunflower oil has been analysed according to the olive oil methods (Line 238 and 240). What reference values have you taken for the parameters K270 and K232? Why were the K values preferred over the peroxide value?
Our reply: As you know, UV spectrum provides repeatable and reproducible indications of the oxidation state of the oils. Conjugated dienes (conjugated hydroperoxides formed in the first oxidation step) absorb at 232 nm, while secondary oxidation products (e.g. unsaturated ketones) absorb at 270 nm. However, conjugated trienes formed during oil decoloration by active earths, also absorb at 270 nm. Thus, in principle, K270 should not be applicable to refined oils. In fact, there are no legal limits for K232 and K270 for refined oils. However, in this study, we decided to use K232 and K270 to assess the oxidation state of the samples after US treatment since all the values were compared to the original sunflower refined oil.
With respect to the use of Peroxide Value, if we consider the instability of peroxides, we may have low PV and a rancid oil due to the decomposition of hydroperoxides into secondary oxidation products (Wolff, 1997). Moreover, the error associated to PV analysis is higher than that for K232 and K270 assays.
Another point that was very important to make our choice is the fact that this study was carried out in the middle of the pandemic situation we are living with periods of complete lockdown. It very much restrained the harvest of the alga on the North of the Portuguese shore, since the work was developed in Lisbon (more than 300 km south). Due to this situation, we had a limited amount of alga to produce enough volumes of the 12 supplemented oil samples for all the different analysis performed in triplicate. For K232 and K270 analysis, we only need a solution of 1 % oil in iso-octane to perform both analyses, while for PV, we need 2-5 g (for PV<12) or 1.2-2.0 g (PV between 12-20). Thus, due to this constraint, we did not have enough oil to perform PV in triplicate, in addition to all other analysis.
Taking into consideration these explanations, we decided to add the following text to the section 3.2.2. Oxidation products (lines 420-428):
“The oxidation state of the oils was evaluated by absorbance measurements at 232 nm and 270 nm, K232 and K270. Primary oxidation products (conjugated dienes, namely conjugated hydroperoxides) absorb at 232 nm, while secondary oxidation products (e.g. unsaturated ketones, short chain fatty acids and aldehydes) absorb at 270 nm. However, conjugated trienes formed during oil refining, in the bleaching process by active earths, also absorb at 270 nm. Thus, in principle, K270 should not be applicable to refined oils (Wolff, 1997). In fact, there are no legal limits for K232 and K270 for refined oils. However, in this study, K232 and K270 values were used to assess the oxidation state of the samples after ultrasound treatment since all the values were compared to the original sunflower refined oil.”
- Sensory evaluation: The reference cited (37) is the ISO 8586 standard, which specifies the criteria for selecting assessors, but does not describe the sensory methodology. The number of judges is small (7), the attributes are not presented and the use of a structured discontinuous scale of only 5 points makes it difficult to detect differences. They should present the deviations of the scores, not only the mean and median.
Our reply: You are right: the reference concerns the general methodology developed for panel selection and training in general. We decided to remove this reference.
In this study, we performed a very preliminary sensory analysis only with the aim of detecting the presence of off-flavours (smell) in supplemented oils. As explained before, we did not have enough oil sample volumes for a more profound sensory analysis. We used assessors for olive oil flavour profile analysis.
According to the EU legislation (COMMISSION REGULATION (EEC) No 2568/91 of 11 July 1991 on the characteristics of olive oil and olive-residue oil and on the relevant methods of analysis), “the panel consists of a panel head and from eight to twelve tasters. However, for the 2001/02 marketing year, the panel may consist of fewer than eight tasters” (ref. 40). According to this legislation, the results are given by the median of the attribute and not by the average value.
Therefore, we think that using 7 judges for this preliminary sensory analysis is sufficient. Moreover, we decided also to indicate the average values, which were not very much different from the median values. Standard deviations for the average values were added to Table 4.
In the text, in the section “2.4.2. Sensory analyses of supplemented sunflower oils”, was rewritten as follows (lines 247-254):
“The supplemented sunflower oil samples were submitted to a preliminary sensory analysis to evaluate the presence of off-odours. A group of 7 trained assessors on olive oil sensory analysis was used [40]. Coded samples were presented to the panellists, in covered glasses at 28-30 °C. They were asked to smell the samples, detect, describe, and quantify the intensity of eventual off-flavours. A discontinuous and structured scale, from 1 (very slight intensity) to 5 (very strong intensity). For all the samples, the mean, and the median (as used for virgin olive oil sensory evaluation) of the detected off-flavours were calculated and used as the response of the panel.”
- Accelerated test (Lines 288-301) I would avoid reference to a method in which time-temperature equivalences are established for migration studies in plastic packaging materials, not for the evolution of the contents in the package. It is sufficient that they indicate that they propose an accelerated study at 60ºC.
Our reply: Thank you for these comments. In fact, this method is used for migration studies in packaging materials. We deleted the text concerning the use of the equation use for migration studies.
- This comment is not relevant to the work, but please review lines 358-363, where phenolic compounds are classified. It is true that the phenolic family is complex and sometimes difficult to classify, but the classification they indicate is wrong. For example, caffeic acid is a water soluble acid and it is hydroxycinnamic.
Our reply: As you suggested, we decided to remove the sentence since it is not relevant to the work.
- Section 3.3: “Pigment content in supplemented sunflower oil” should not contain the subsection Flavonoids (3.3.3). Flavonoids should be presented together with phenols (3.5).
Our reply: Thank you for this suggestion. We moved it to the section 3.5. In the first version of the manuscript, we had decided to present flavonoids together with the other pigments because they are also pigments. Due to this change, the results concerning flavonoids were deleted from Table 5, where they were presented together with the chlorophyll pigments and carotenoids, and put in Table 6, together with phenolic contents. A new Table (Table 7) was created to present the results about antioxidant activity assayed by DPPH and FRAP.
Following this rational, we decided to delete the results concerning flavonoids in alga extracts from section “3.1.2. Chlorophyll and carotenoid pigment contents” to section 3.1.3. The flavonoid contents were moved from Table 2 to Table 3.
- If 9 to 12 treatment are control, how do you justify the internal differences between the control treatments and how do they affect the outcome of the other treatments?
Our reply: Concerning treatments 9 to 12, they are not control. They are the central point experiments. The variability observed in these repetitions is used to calculate the error of the polynomial models describing the response surfaces fitted to the experimental points. The internal differences come from the variability in alga and to experimental errors. This is explained in lines 220-225.
- What is the purpose of expressing flavonoids such as both flavonoids, quercetin and catechin?
Our reply: Quercetin and catechin are the flavonoids that are the most widely distributed in edible plants. On the other hand, several authors such as Saad et al., (2016), refer the presence of catechin and quercetin in algae (Red Marine Alga Alsidium corallinum, DOI: 10.1002/tox.22368). In this work, the authors expressed flavonoids using more than one standard to make comparison to other authors easier, as several are used in the literature (including rutin, beyond quercetin and catechin).
The following text was included in the section 3.5.2. Flavonoids:
"Quercetin and catechin are the flavonoids that are the most widely distributed in edible plants. Moreover, several authors, such as Saad et al. [16], refer the presence of catechin and quercetin in algae (Red Marine Alga Alsidium corallinum). In this work, the authors expressed flavonoids using more than one standard to make comparison to other authors easier, as several are used in the literature (including rutin, beyond quercetin and catechin).
- Lines 419 to 424: I consider this paragraph elucubrative and unjustified. The unpleasant taste and off-flavour of algae is related to minerality, amino acids, free polyunsaturated fatty acids and some volatile compounds from secondary decomposition of hydroxyperoxides and other pathways ... If the judges are sufficiently trained they should differentiate between positive and negative attributes due to the presence of seaweed. The aroma of algae in an algae infused oil should not be a defect to a trained panel of judges. For future studies I recommend that the expert assessors analyse the intensity of the attributes of a sensory profile and an affective panel indicate their degree of satisfaction.
Our reply: The judges considered the smell of the supplemented oils as having negative attributes described as “alga/fishy/rotten fish/ammonium”. Alga smell was not considered as a pleasant attribute in supplemented sunflower refined oils. No positive attributes (smell) coming from the seaweeds were detected.
Thank you for your recommendation for future studies. As explained before, we could not perform a complete flavour profile analysis (15 mL/sample x 10 assessors= 150 mL minimum). For the same reason, affective evaluation was even more difficult to carry out: 60 consumers x 15 mL sample (900 mL minimum of each sample). We hope to be able to perform these sensory evaluations in a near future.
- In the tables 4, 5 and 6 there is no statistic parameter expressing the deviation.
Our reply: The STD were added to the tables. Extraction yields were calculated from the average value of each assay. Thus, no STD were added to the yields.
- The authors say: “Therefore, it would be necessary to refine (deodorize) the supplemented oils, to remove these off-flavours”. I do not believe that deodorization is the solution, as it can cause unwanted thermal effects. There are research groups working on encapsulating algae derivatives to attenuate odour when used in preparations for which odour is not acceptable.
Our reply: The authors understand the reviewers’ point of view. However, the aim of this study was to enhance the nutritional quality and oxidative stability of refined sunflower oil by supplementation with bioactive compounds, using a direct extraction from algae into the oil. In this case, the encapsulation of algae derivatives would not be feasible.
Deodorisation is one of the steps of oil refining. It is by refining under optimized and controlled conditions that crude oils are transformed into edible oils. Refining is not a harmful process. Therefore, from an industrial point of view, we believe it is important to keep the previous sentence.
- Line 322: Please, change oxidative stability by oxidative status.
Our reply: We could not find “oxidative stability” in the line 322 but in the section 3.2.2. Oxidation products, line 404, where it seems appropriate the change you suggested. We did it.
- Line 662: Please, change oxidative stability by antioxidant activity.
Our reply: Thank you for this suggestion. However, we could not find this expression in line 662 but in line 668. We found appropriate to change oxidative stability by antioxidant in line 668.
- For further research I recommend reviewing the antioxidant methods selected and including some method of lipid peroxidation (TBARS, control of hexanal and other end-products)
Our reply: The authors thank for your recommendation for future studies.
Best regards,
Suzana Ferreira-Dias

Reviewer 2 Report
This study aimed to enhace the nutritional quality and oxidative stability of refined sunflower oil by supplementation with bioactive compounds, mainly antioxidants, from P. canaliculata directly to the oil by ultrasound assisted extraction. The effect of the extraction time and concentration of algae on oil oxidative stability, as well as on the content of bioactive compounds, were evaluated by response surface methodology, following a central composite rotatable design. In addition, the oxidative stability of supplemented sunflower oil, obtained under selected extraction conditions, was also evaluated at 60 °C. The manuscript is well-planned and written, the methods are good explained and the results clearly discussed. For all these reasons I recommend to publish this work with minor modifications.
- Line 24: abbreviatures explained the first time (DPPH, FRAP)
- Lines 70-7: "This alga is rich (...) phenolic compounds" - This statment needs a reference.
- Lines 513, 515, 592, 624: G. changii after first time.
- Lines 704, 705: Caption of figure 4 is not clear
- About references, only 15 of 59 references are from last 5 years. I recommend to include more new literature.
Author Response
Reviewer comments and our replies
Reviewer 2
We thank the Reviewer #2 for the supportive comments, and for the time and effort that was obviously put in here to improve our manuscript.
We believe we have integrated all the comments of Reviewer #2 listed below into this revision.
Comments and Suggestions for Authors
This study aimed to enhance the nutritional quality and oxidative stability of refined sunflower oil by supplementation with bioactive compounds, mainly antioxidants, from P. canaliculata directly to the oil by ultrasound assisted extraction. The effect of the extraction time and concentration of algae on oil oxidative stability, as well as on the content of bioactive compounds, were evaluated by response surface methodology, following a central composite rotatable design. In addition, the oxidative stability of supplemented sunflower oil, obtained under selected extraction conditions, was also evaluated at 60 °C. The manuscript is well-planned and written, the methods are good explained and the results clearly discussed. For all these reasons I recommend to publish this work with minor modifications.
- Line 24: abbreviatures explained the first time (DPPH, FRAP)
Our reply: We added DPPH (2,2-diphenyl-1-picrylhydrazyl) and FRAP (Ferric Reducing Antioxidant Power) to the abstract, as suggested.
- Lines 70-7: "This alga is rich (...) phenolic compounds" - This statment needs a reference.
Our reply: As you suggested, the following references were added [22-25]:
- Chater, P.I.; Wilcox, M.; Cherry, P.; Herford, A.; Mustar, S.; Wheater, H.; Brownlee, I.; Seal, C.; Pearson, J. Inhibitory activity of extracts of Hebridean brown seaweeds on lipase activity. J Appl Phycol 2016, 28, 1303-1313.
- Cherry, P.; O’Hara, C.; Magee, P.J.; McSorley, E.M.; Allsopp, P.J. Risks and benefits of consuming edible seaweeds. Nutr Rev 2019, 77, 307-329.
- Colliec, S.; Boisson-vidal, C.; Jozefonvicz, J. A low molecular weight fucoidan fraction from the brown seaweed Pelvetia canaliculata. Phytochemistry 1994, 35, 697-700.
- Makkar, H.P.; Tran, G.; Heuzé, V.; Giger-Reverdin, S.; Lessire, M.; Lebas, F.; Ankers, P. Seaweeds for livestock diets: A review, Anim Feed Sci Tech 2016, 212, 1-17.
- Lines 513, 515, 592, 624: G. changii after first time.
Our reply: done
- Lines 704, 705: Caption of figure 4 is not clear
Our reply: The caption was changed as follows:
Figure 4. Accelerated oxidation tests carried out under dark at 60 ˚C, for 12 days, of non-supplemented and supplemented sunflower oil obtained by UAE with Pelvetia (12.5 %, m/v) for 20 min. The antioxidant activity was assayed by (a) DPPH and (b) FRAP methods
- About references, only 15 of 59 references are from last 5 years. I recommend to include more new literature.
Our reply: Thank you for your recommendation, as you suggest we add several reference from the last 5 years, as follows:
- Aminzare, M.; Hashemi, M.; Ansarian, E.; Bimkar, M.; Azar, H.H.; Mehrasbi, M.R.; Daneshamooz, S.; Raeisi, M.; Jannat, B.; Afshari, A. Using natural antioxidants in meat and meat products as preservatives: A review, Adv. Anim. Vet. Sci. 2019, 7, 417-426.
- Kirke, D.A.; Smyth, T.J.; Rai, D.K.; Kenny, O.; Stengel, D.B. The chemical and antioxidant stability of isolated low molecular weight phlorotannins. Food chem 2017, 221, 1104-1112.
- Makkar, H.P.; Tran, G.; Heuzé, V.; Giger-Reverdin, S.; Lessire, M.; Lebas, F.; Ankers, P. Seaweeds for livestock diets: A review, Anim Feed Sci Tech 2016, 212, 1-17.
- Zhang, R.; Yuen, A.K.; Magnusson, M.; Wright, J.T.; de Nys, R.; Masters, A.F.; Maschmeyer, T. A comparative assessment of the activity and structure of phlorotannins from the brown seaweed Carpophyllum flexuosum. Algal Res. 2018, 29, 130-141.
- Ben Saad, H.; Gargouri, M.; Kallel, F.; Chaabene, R.; Boudawara, T.; Jamoussi, K.;Magné, C.; Zeghal, K.M.; Hakim, A.; Ben Amara, I. Flavonoid compounds from the red marine alga Alsidium corallinum protect against potassium bromate‐induced nephrotoxicity in adult mice. Environ. Toxicol 2016, 32, 1475-1486.
- Kubalt, K. The role of phenolic compounds in plant resistance. Biotechnol Food Sci 2016, 80, 97-108
Best regards,
Suzana Ferreira-Dias
Round 2
Reviewer 1 Report
Suggestions to organize the presentation of results and the modification of some parts of the text have been considered.
I appreciate the effort to improve the manuscript.